# Effect of Freeze-Dried Camel Rennet Extract on Coagulation of Camel–Goat Milk Mixture and Characterization of the Cheese Obtained

**DOI:** 10.3390/foods14030334

**Published:** 2025-01-21

**Authors:** Biya Bouras, Ouarda Aissaoui-Zitoun, Férial Aziza Benyahia, Souhila Djema, Leila Bouras, Mohammed Nassereddine Zidoune, Imène Felfoul

**Affiliations:** 1Research Laboratory of Biology, Environment and Health (LBEH), University of El Oued, El Oued 39000, Algeria; 2Laboratory of Nutrition and Food Technology (LNTA), Institute of Nutrition, Feeding and Agrofood Technology (INATAA), University Mentouri Brothers Constantine 1, Constantine 25017, Algeria; azouarda@yahoo.fr (O.A.-Z.); zidounem@yahoo.fr (M.N.Z.); 3Agro-Food Engineering Laboratory (GENIAAL), Institute of Nutrition, Feeding and Agrofood Technology (INATAA), University Mentouri Brothers Constantine 1, Constantine 25017, Algeria; ferialaziza@yahoo.fr; 4Research Center in Physicochemical Analysis (CRAPC), Bou Ismail 42004, Algeria; souhiladjemahadidi@gmail.com; 5Research Laboratory of Exploitation and Valorization of Saharan Energy Resources (EVSER), University of El Oued, El Oued 39000, Algeria; leila.brs.tec@gmail.com; 6Valuation, Analysis and Food Safety Laboratory (LAVASA), National Engineering School of Sfax (ENIS), University of Sfax, Route Soukra, Sfax 3038, Tunisia

**Keywords:** freeze-dried camel rennet, camel milk, goat milk, coagulation, cheese

## Abstract

This study aims at the use of freeze-dried camel rennet extract (FDCR) in the manufacture of fresh cheeses from a mixture of camel and goat milk in comparison with the microbial coagulating agent (FDMC). Physical properties, chemical composition, microstructure, and sensory analysis of the cheeses were performed. The recommended amount of FDCR for coagulation of camel–goat milk mixture was 0.2 g/L. The cheese obtained was mainly characterized by dry matter 34.99 ± 0.57% and protein content 36.26 ± 1.75%/DM. Texture profile analysis revealed that the obtained cheese was mainly characterized by cohesiveness 0.32 ± 0.01 and springiness 14.25 ± 0.63 mm. The microstructure showed that the obtained cheese had more and wider pores. FTIR was used to monitor the differences in the gross composition of the obtained cheese compared to that coagulated with FDMC. The main difference was the presence of amide I in the cheese coagulated with FDCR. For X-ray diffraction, the results noted that the use of FDCR as a natural extract in the coagulation of camel–goat milk mixture did not lead to the appearance of crystalline structure in the cheese. For sensory evaluation, the panelists preferred the cheese coagulated with FDCR with a score of 9/15.

## 1. Introduction

Camels are kept by pastoralists’ subsistence production systems mainly for their milk production. They are well known for maintaining milk production during drought conditions. In drought-stricken areas of the world, where continued drought decimates cattle, sheep, and goat populations, only camels survive and continue to produce milk [1]. In Algeria, the camel is one of the greatest reservoirs of wealth and resources in the Saharan territory [2]. In fact, the Algerian camel population is estimated at 417,322 heads in 2020, representing 1.12% of the world population, 1.19% of the African population, and 7.48% of the Maghreb population, with a mean annual growth corresponding to 3% recorded between 1961 and 2018, and a milk volume of the 15080 tons produced in 2020 [3]. In addition, according to the FAO (Food and Agriculture Organization), the goat population in Algeria was estimated at 4.9 million in 2018 [4]. The nutritional and digestive qualities of goat milk are undeniable, i.e., it is less allergenic, undergoes lactic fermentation more slowly than cow milk, and it is used as a medicine for the treatment of certain diseases [5].

The production of cheese from camel milk has always been a challenge due to difficulties in milk coagulation because of the lack of κ-casein and its specific enzyme cleavage site [6]. The casein in camel milk is characterized by a larger micelle size, with an average diameter of 380 nm, in contrast to 150 nm, 260 nm, and 180 nm observed in cow, caprine, and ovine milk, respectively [7,8]. Wangoh et al. [9] carried out the first research on the coagulation of camel milk using enzymatic extracts from the abomasum of camels.

Recently, Lambé et al. [10] and according to the recommendations of the European Commission, reported that the use of the coagulating enzyme extracted from the abomasum of ruminants does not present toxicological risks or allergenic reactions for consumers.

For the production of mixed milk cheese, camel milk can be used with sheep milk [11], buffalo milk [12], and cow milk [13]. Aissaoui-Zitoun et al. [14] and Bouras et al. [15,16] initiate the research work carried out on the study of the enzymatic coagulation of camel milk combined with goat milk and on the quality of the cheese made, respectively, with liquid chicken pepsin and liquid camel rennet. Moreover, Khan et al. [17] reported that the addition of starter cultures for the manufacture of fresh camel milk cheese reduced the coagulation time of camel milk.

To the authors’ knowledge, only a few studies in the literature have investigated the use of freeze-dried camel rennet extract in the coagulation of camel milk. These studies reported that freeze-dried rennet extracts were used in camel milk coagulation but not in the camel–goat mixture. In Algeria, camel breeding is sometimes integrated with goat breeding and some breeders mix the milk from their camels with that of goat milk to produce butter and fresh cheese [18]. In this context, the aim of this study comparing two coagulation methods in the process of making fresh cheeses from a mixture of camel milk and goat milk, as well as the physicochemical, mineral, sensory and microstructural characterization of the obtained cheeses.

## 2. Materials and Methods

### 2.1. Milk Samples

Camel and goat milk was collected, respectively, from four dromedaries of the *Sahrawi* population (*Camelus dromedarius*) and three goats of the *Arbia* population from local farms in the South East of Algeria (El Oued region). The lactation time was 30 days for camels, while goats were in lactation for 22 days. The farming method is semi-intensive for camels and intensive for goats. The camel and goat milk was collected cleanly in sterile milking cans and sent to a laboratory with a cold chain using ice packs at 4 °C within 45 min. Each type of milk was collected alone; the mixture of camel milk and goat milk was carried out in the laboratory after measuring the pH. Then, the mixed milk was kept at 4 °C (RH~75%) in autoclaved bottles. The mixed milk samples were used on the same day of collection for all studied tests.

### 2.2. Coagulant and Starter Cultures

#### 2.2.1. Camel Abomasum

The last part of the third compartment (L3C) was collected in an approved slaughterhouse from a controlled and non-sick dromedary (aged less than one year) from the El Oued region in Algeria. The L3C was washed in cold water then frozen at −18 °C until use. The camel slaughterhouse practices were controlled by a state-approved veterinarian and under ethical slaughter authorization.

#### 2.2.2. Freeze-Dried Microbial Coagulant

The microbial coagulant is a commercial coagulant of fungal origin. It is obtained from *Rhizomucor miehei* (Marzyme R 150MG) with a coagulating force of 1/5000. The product was purchased from DANISCO (reference n°53857). The freeze-dried microbial coagulant (FDMC) was obtained in the laboratory using a freeze dryer (Grosseron, CHRIST Alpha 1–2 LSC Basic, Couëron, Germany) under −50 °C and 0.040 mbar for comparison with the freeze-dried camel rennet extract.

#### 2.2.3. Mesophilic and Thermophilic Commercial Starter Cultures

The mesophilic starter cultures (MSC) used in the inoculation of the camel and goat milk mixture produces the aroma and CO_2_ of *Lactococcus lactis* subsp*. cremoris*, *Leuconostoc lactis*, *Lactococcus lactis* subsp*, lactococcus lactis* subsp. *biavar diacetylactis*. Thermophilic starter cultures (TSC) of yogurt are *Lactobacillus bulgaricus* and *Streptoccocus thermophilus*. The two types of commercial ferments (FD-DVS FLORA DANICA, CHR HANSEN) were purchased from Novonesis Group- Food & Beverage Biosolutions, Jernholmen, Denmark.

### 2.3. Extraction and Freeze-Drying of Camel Rennet

The extraction of camel rennet was carried out according to the protocol used by Wangoh et al. [9]. The L3C was sliced (1 cm^2^) and then macerated in 6% NaCl solution (1:10 *w*/*v*) containing 2% boric acid continuously for 4 days at 5 °C. Then, the mixture was filtered and centrifuged at 1500 rpm for 15 min at 25 °C. The pH of the supernatant was decreased from 5.5 to 4.7 with HCl (1 N) and the extracts were kept at 25 °C for 24 h to activate the zymogens. The pH was then raised to 5.5 with NaOH (1 N). After centrifugation at 1500 rpm for 15 min at 25 °C, the final rennet extract was obtained. Finally, the extract of the final camel rennet was freeze-dried (FDCR) under the same conditions as the microbial coagulating agent (Section 2.2.2).

### 2.4. Acidification Activity of Commercial Starter Cultures

In order to determine the acidification activity of the blend between mesophilic and thermophilic starter cultures (BMTSC) before the manufacture of fresh cheeses, the acidification activity was tested in a mixture of camel and goat milk. The mixtures prepared are as follows:

M_1_: Camel and goat milk mixture + inoculum of MSC;

M_2_: Camel and goat milk mixture + inoculum of TSC;

M_3_: Camel and goat milk mixture + inoculum of BMTSC.

The inoculum of each type of ferment was prepared by adding 1 U of MSC or TSC to 100 mL of camel and goat milk mixture. Regarding the inoculum of the blend of the two ferments, 1 U of MSC and 1 U of TSC were added to the same volume of the mixed milk. Then, pH was measured using a pH meter (inolab, WTW™ 7110, Boulevard Sebastien Brant, France) of M_1_, M_2_, and M_3_ every 4 h during a 24 h period. The temperature was set, respectively, at 30, 40, and 35 °C for the inoculum of MSC, TSC and the blend of the two ferments and the fermentation lasted 24 h.

### 2.5. Optimization of Enzymatic Coagulation

In order to optimize the amount of FDCR and FDMC (enzymatic clotting), the yield of cheese prepared with a mixture of camel and goat milk was determined using the following doses: 0.08, 0.12, 0.16, 0.20, 0.24, and 0.28 g/L. The optimized amount to be used in the manufacture of fresh cheese was chosen based on two essential recommended visual criterion: (1) the coagulation time should not exceed 30 min, (2) the fresh cheese should be characterized by a firm and unbroken coagulum. In addition, we recommended a textural criterion: (3) the coagulum should have a non-sticky texture on the touch after draining. Cheese yield (Equation (1)) is defined as the percentage of total cheese weight (Kg) relative to the initial weight of milk (Kg) [19]. The experiments were carried out in triplicate.
Cheese yield (%) = (Cheese weight/Milk weight) × 100(1)

### 2.6. Cheese-Making Process of Camel and Goat Milk Mixture

Raw camel and goat milk was pasteurized at 63 °C for 30 min. After cooling to 4 °C, the milk was mixed at a ratio of 50%/50% (*v*/*v*), and this mixture was used in the manufacture of fresh cheese. Then, the pasteurized camel and goat milk mixture (1 L for each test) was filtered through a sieve with a drainage hole diameter of 2 mm. The mixed milk was sieved at a 50-µm mesh size (test sieve, Mesh S-Steel, 200 mm diameter, 50 mm height, Retsch GmbH, Haan, Germany) in order to remove the potential residual coalesced fat globules. Then, an amount of 0.015% of calcium chloride (CaCl_2_) was added. In addition, the milk was inoculated with 20 mL/L of BMTSC inoculum, then the mixture was maintained for 30 min at 35 °C. After that, the water bath was set to 37.75 and 38.84 °C for FDCR and FDMC, respectively. In parallel, the pH of the mixed milk was adjusted with lactic acid to 6.16 and 6.31, for FDCR and FDMC, respectively [16]. Then, 7 g/L of NaCl and 0.2 g/L of FDCR and FDMC were added. The coagulating strength was, respectively, 1/52038 SU and 1/53262 SU for FDMC and FDMC. After coagulation, the cheese was spontaneously drained and molded without cutting. Finally, the cheeses coagulated with FDCR (CC) and FDMC (MC) were placed in clean glass food containers and kept at 4 °C (RH~75%) for 24 h before the analyses. The glass food containers were covered with cling film to prevent water evaporation.

### 2.7. Study of CC and MC Fresh Cheeses

#### 2.7.1. Color Measurement

The measurement of color (*L**, *a**, *b**, C*, *h*°) of CC and MC fresh cheeses stored at 4 °C was carried out using a colorimeter (Konica Minolta, Inc, Japan) [20]. The chroma (*C**) and hue angle (*h*°) indicating, respectively, the saturation level and the shade of the color. *C** and *h*° are calculated according to Equations (2) and (3) as follow:(2)$$C^{*}=\sqrt{\left(a^{*2}+b^{*2}\right)}$$
(3)$$h^{{}^{\circ}}=\arctan\left(b/a\right)$$*L** value indicates sample lightness from black (0) to white (100), *a** value signifies color varying from redness (+) to greenness (−), and *b** value varies from yellowness (+) to blueness (−).

#### 2.7.2. Crude Mineral Composition and Cheese Yield Measurement

The crude composition of CC and MC cheese is carried out according to AOAC [21]. pH is determined by dissolving 1 g of newly made fresh cheese in 10 mL of distilled water using a digital pH meter (Inolab, Weilheim, Germany). The dry matter content is determined by drying 3 g of fresh cheese in an oven (Binder, Halvatia, France) at 105 °C for 24 h. Ash content is determined by total incineration of the dry material at 550 °C for 7 h. The total nitrogen content was carried out according to Kjeldahl method (Buchi, Easy Kjel, Couëron, France) and protein content was deduced from total nitrogen content by multiplying by a conversion factor of 6.38. Fat content was determined according to the Van Gulik method (Funke Gerber centrifugal, Berlin, Germany). The mineral elements (Ca, P, Na, K, Mg, Fe, Mn, and Cu) were measured separately using an atomic absorption spectrophotometer (Thermo scientific, ICE 3000, Norwalk, CA, USA). The measurement of physicochemical parameters and minerals was carried out in triplicate.

#### 2.7.3. Water Activity Measurement

The water activity (aw) of fresh cheese was determined according to ISO 21807 [22] using a water activity meter (AW SPRINT TH500, Novasina, Lachen, Switzerland) in quick mode at 25 °C. The a_w_ measurement is triplicated.

#### 2.7.4. Texture Profile Analysis

Uniaxial compression was realized through Textural Profile Analysis (TPA) test (Texture Analyzer, LLOYD instruments, Fareham, UK). Cylindrical CC and MC fresh cheese samples of 10 mm in diameter, 40 mm in height, and 45 mm in width were made by perforating the cheese samples vertically at room temperature (20 °C) and then analyzed immediately. The compression test was carried out with a cylindrical probe of 25 mm diameter at a speed of 0.83 mm/s by compressing the sample to 70% of its initial height. The probe of compression was lubricated using low-viscosity oil between each analysis to eliminate the frictional effects during the uniaxial compression. All operations were conducted through software provided by Texture Technologies Corp connected to the instrument [23]. Texture profile attributes including hardness (N), adhesiveness (N), cohesiveness, springiness (mm), and chewability (Nmm) were analyzed in triplicate.

#### 2.7.5. Fourier Transform InfraRed Spectroscopy Measurement

FTIR spectra of CC and MC fresh cheeses were obtained using FTIR (Agilent Technologies, Carry 630, Santa Clara, CA, USA). A slice of each type of cheese was placed directly on top of the crystal and either pressed down onto or measured without pressing. The spectra were recorded at 20 °C from 4000 to 450 cm^−1^ at a normal resolution of 8 cm^−1^ with an accumulation of four scans for each spectrum. The background spectrum, which contains the absorptions of molecules present in the air, was scanned at the beginning of measurement by pouring distilled water over the ATR cell. The same procedure was used for scanning the blank spectrum, which contained the absorption of distilled water. After each measurement, the ATR crystal was thoroughly washed with ethanol and distilled water and then dried. The data were detected in the transmission mode [24,25]. For each experiment, three replicates were performed.

#### 2.7.6. X-Ray Diffraction Measurement

The XRD profiles of CC and MC fresh cheeses were investigated using powder X-ray Diffractometer (Proto Manufacturing Inc., Taylor, MI, USA) The theta angle was adjusted, ranging from 5° to 50°, while the scanning rate remained constant at 1°/s. Operating voltage was 40 KV [25].

#### 2.7.7. Microstructure Under Environmental Scanning Electron Microscope

A small piece of each of the CC and MC fresh cheeses was finely cut (0.5 cm in length and 0.5 cm in width) and air-dried for 4 to 5 h. Then, the cheese samples were dried in an atmosphere saturated with glutaraldehyde overnight (12 h) at 4 °C. Then, each piece of cheese was fixed by a series of ethyl alcohol from 10° to 100° for 5 min in solution. Subsequently, the pieces of cheese were dried in the open air for additional hours before proceeding to observation. The latter was measured under an environmental scanning electron microscope (FEI Quanta 250 FEG, Hilsboro, OR, USA) operating under a large file detector (LFD) and low vacuum with an accelerating voltage of 1000 kV [26]. The processing of micrographs was carried out with the software image J Fiji v1.54g USA.

#### 2.7.8. Sensory Analysis

Sensory analysis was carried out using 20-panel members between students and teachers. The panelists were trained in advance for two days. They were formed for every descriptive vocabulary of the basic third sensory modalities, that is, appearance, texture, odor, and taste. The cheeses were provided to the panelists (10 g of each type of cheese for each panelist). Then, the panelists were invited to scale each descriptor’s intensity on a 15-point scale. The cheeses were coded (three-digit code) and randomly provided to the panelists [27]. The evaluation of the CC and MC fresh cheese products was carried out independently at room temperature (20 °C) in triplicate.

### 2.8. Statistical Analysis

All experiments and analyses were carried out in triplicates and the obtained data were expressed as means ± standard deviation. Statistical analysis was performed using Minitab 19 software (Minitab Inc., State College, PA, USA) using Student’s *t*-test with α = 95%.

## 3. Results and Discussions

### 3.1. Starter Culture Performance in Camel–Goat Milk Mixture

The acidification kinetics of the camel–goat milk mixture using different starter cultures are shown in Figure 1.

The initial pH of M_1_, M_2_, and M_3_ was 6.60 ± 0.00. A slight decrease in pH was recorded after 4 h of inoculation. After 8 h, the acidification of milk was accelerated with the use of the M_3_ mixture, containing the blend of MSC and TSC. Moreover, similar acidification kinetics were obtained for both M_1_ and M_2_ mixtures, corresponding to MSC and TSC, respectively. Indeed, the increase in lactic acidity occurred almost in the same way for both samples. It is important to note that the final pH values after 24 h fermentation were 4.80 ± 0.01 for M_1_, 5.10 ± 0.00 for M_2_, and 4.30 ± 0.02 for M_3_. Based on the results obtained from the acidification kinetics of the starter cultures studied, the use of BMTSC (M_3_) for the manufacture of fresh cheese inoculated with the mixture of camel and goat milk could be the best choice, since this mixture showed faster acidification compared to other starter culture mixtures (M_1_ and M_2_).

Furthermore, Bekele et al. [28] studied the level of acidification of pasteurized camel milk by different commercial starter cultures. The authors reported a minimum pH of 6.24 ± 0.00 after 70 h of fermentation. Al-zoreky and Almathen [29] investigated the use of thermophilic yogurt culture for camel milk acidification. They found that the acidification level was about 5.27 ± 0.05. Abbaschian et al. [30] reported that the curd formation depends on the rate of acidification and the use of the thermophilic/mesophilic starter culture mixture is recommended for camel milk coagulation. Based on the results obtained in the present study, it can be deduced that the addition of goat milk to camel milk in equal volume influenced the rate of acidification compared to the results previously reported by Bekele et al., Al-zoreky, and Almathen [28,29] for the acidification of camel milk without the addition of other types of milk. The fortification of camel milk with other types of milk improves acidification since camel milk has a weak buffering capacity [31].

Moreover, the rate of acidification obtained was further related to the use of the blend of mesophilic and thermophilic starter culture and its added concentration in mixed milk. Recently, Quadeer et al. [32] studied the influence of thermophilic and mesophilic starter culture in mixed milk. They reported that the acidity was highest with 0.83% of camel milk added to 10% of buffalo milk and fermented with thermophilic starter culture. Seifu [6] reported that the type and the amount of starter culture affects the quality of camel milk cheese.

### 3.2. Cheese Yields

The results of the cheese yields of the different trials using FDCR and FDMC are presented in Table 1. The results obtained showed a significant difference (*p* < 0.05) between the cheese yields recorded for the fresh cheeses produced with FDCR and FDMC in different amounts.

CY_1_ values vary from 15.88 ± 0.12% (minimum yield) recorded with 0.12 g of FDCR to 19.28 ± 0.15% (maximum yield) recorded with 0.16 g of FDCR. A minimum yield of 9.79 ± 0.14% was obtained with 0.08 g of FDMC and a maximum yield of 14.92 ± 0.07% with 0.2 g. From these results, it can be noted that increasing the dose of FDCR and FDMC for the coagulation of pasteurized camel–goat milk mixture does not lead to an increase in the cheese yield. Moreover, all the values of the cheese yield recorded of CY_1_ were significantly higher than those of CY_2_ at different amounts of FDCR and FDMC.

Bouras et al. [16] reported a cheese yield of 17 ± 1.8% for camel–goat milk mixture coagulated with the liquid camel rennet extract and 18.3 ± 0.4% with the liquid microbial coagulating agent. For comparison with these results, FDCR is recommended, and its use is cost-effective in the manufacture of goat or camel milk cheese. In terms of other milk blends, Gemechu et al. [33] reported fresh cheese yield of 21.18 ± 1.26% for the blend of 50% camel milk and 50% cow milk. Shahein et al. [10] reported cheese yields between 14.7% and 20.1% for different blends of camel milk with buffalo milk.

The different images of cheeses produced with freeze-dried camel rennet (CC) and freeze-dried microbial coagulant (MC) are shown in Figure 1. The consistency of the coagulum varies according to the amount added of FDCR and FDMC from 0.08 g/L to 0.28 g/L. Regarding the clotting time, a minimum time of 5 ± 0.00 s to 43 ± 0.00 min was recorded with FDCR. For FDMC, the clotting time ranged from 3 ± 0.00 s to 45 ± 0.01 min. According to visual and textural criteria, the enzymatic coagulation of pasteurized camel–goat milk mixture was optimal at 0.2 g/L of FDCR and FDMC. As a result, this quantity was chosen for the production of the finished fresh cheeses.

### 3.3. Color Characteristics of Fresh Cheeses

Table 2 displays the color parameters of CC and MC fresh cheeses. The color of cheese represents a quality attribute that influences customer preferences and purchasing decisions. It is important to point out that the use of FDCR had a significant effect (*p* < 0.05) on color parameters compared to cheese produced with FDMC. The *L** and *a** values of MC cheese (93.97 ± 0.03/−1.75 ± 0.01) were slightly, but significantly, higher than those recorded with CC cheese (92.86 ± 0.23/−1.03 ± 0.02). In addition, the value of *b** of CC cheese (11.78 ± 0.08) was slightly higher than of MC cheese (10.33 ± 0.11). These differences might be attributed to the brownish color of FDCR compared to FDMC, which is white. Mbye et al. [34] reported values between (−0.9 ± 0.01/−1.7 ± 0.03) for *a**, and between (5.8 ± 0.05/14.7 ± 0.31) for *b** for different camel cheeses. These results were linked to the use of camel chymosin and *Withania coagulans* extract.

In addition, Kaczyński et al. [35] reported that the *L** value decreased with increasing ripening time and the use of whole milk in cheese-making. In addition, the presence of a high moisture content leads to a high value of *L**. In our study, these results were linked with the mixture of camel and goat milk in equal volumes.

### 3.4. Crude and Mineral Composition of Fresh Cheeses

The crude and mineral composition of CC and MC cheeses is displayed in Table 2. A significant difference (*p* < 0.05) was recorded between the values of the physicochemical parameters of CC and MC cheeses. Indeed, both cheeses were slightly acidic with a pH value below 5 with a significantly lower pH value reported for CC cheese compared to MC cheese (4.50 ± 0.00 vs. 4.66 ± 0.00, respectively). The CC and MC cheeses contained high a_w_ values of 0.96 ± 0.00 and 0.97 ± 0.00, respectively. As shown in Table 2, protein content was significantly higher for CC cheese compared to MC cheese, which explains that the freeze-drying of the enzymatic extract of camel rennet (FDCR) had no significant effect on the coagulation of the goat-camel milk mixture. Based on MFFB, CC and MC cheeses could be characterized as soft cheeses (with moisture content > 67%) [36]. The use of FDCR for the coagulation of the goat–camel milk mixture gave a significantly higher cheese yield than FDMC (15.96 ± 0.10% vs. 13.55 ± 0.01%, respectively).

Bouras et al. [16] investigated the physicochemical characteristics of fresh cheese made from a mixture of camel and goat milk and coagulated with non-freeze-dried camel rennet extract. These authors reported that the dry matter content was higher than that obtained in the present study (26.70 ± 2.10%). This difference could be attributed to the total dry matter content of the mixed milk and/or the coagulant activity of FDCR.

Ash content presented a significant difference (*p* < 0.05) between CC and MC cheeses with 1.41 ± 0.02 and 3.55± 0.00%, respectively. Indeed, a high sodium content was recorded in both cheeses with 123.66 ± 0.01 mg/100 g for CC cheese and 128.66 ± 0.00 mg/100 g for MC cheese. These results could be explained by the salinity of camel milk from the El Oued region [37]. Calcium and magnesium contents in CC cheese (4.33 ± 0.02 and 3.00 ± 0.00 mg/100 g) were significantly higher compared to MC cheese (1.33 ± 0.01 and 1.33 ± 0.01 mg/100 g). On the other hand, no significant difference was recorded between both cheeses regarding phosphorus, iron, manganese, and copper contents.

It is important to point out that only a few studies have focused on making fresh cheese from a mixture of camel–goat milk, thus the interpretation of the results seems difficult. At this level, it can be assumed that the physicochemical and mineral characteristics are closely linked to the origin of the milks, the type of coagulant and the lactic ferments used.

### 3.5. Texture Parameters of Fresh Cheeses

Table 2 presented the outcomes of texture profile parameters (hardness, cohesiveness, springiness, adhesiveness, and chewiness) of CC and MC cheeses. No significant difference was observed for hardness between CC and MC cheeses (*p* < 0.05). This could be explained by the fact that both cheeses are fresh and underwent the same manufacturing process and draining time. On the other hand, significant differences (*p* < 0.05) were recorded for cohesiveness, springiness, adhesiveness, and chewiness. Camel milk cheese is characterized by a less hard texture compared to cow milk cheese; due to the low casein content in camel milk (60% of the total protein) compared to 80% in cow milk [38]. Mekkaoui et al. [39] studied the variation in the texture of fresh camel milk cheese coagulated with commercial chymosin; depending on the breeding of camels (*Camelus dromedarius*) in extensive and intensive systems. They noted that camel milk cheese from dromedaries in extensive breeding system appears to have a higher hardness than that from dromedaries in a semi-intensive system with, respectively, 1.85 ± 0.35 and 1.14 ± 0.33 N. Furthermore, the texture of fresh camel cheese was affected by the nature of the starter culture [28] as well as the concentration of calf rennet [40] and cold storage [41]. These last authors recorded the values of hardness (2.136 N), cohesiveness (0.207), springiness (0.275 mm), gumminess (0.637 N), and chewiness (0.216 Nmm). In the present study, a high chewiness value was obtained for CC (7.12 ± 1.44) and MC (14.80 ± 1.04) fresh cheeses.

Recently, Mbye et al. [42] studied the textural characteristics of ultrafiltered and heat-processed camel milk fresh cheeses. They reported that the hardness and chewiness of the cheeses increased with increases to the concentration of ultrafiltration of camel milk from 1- to 2-fold. In addition, ultrafiltrated treated camel cheeses may also have higher hardnesses due to a higher total solid content of the retentate and higher calcium retention in the milk. In terms of blended milk, Boukria et al. [43] studied the texture of cow milk and camel milk gel after coagulation. They reported that the addition of camel milk to more than 50% increased the weakness of the gel protein network. Moreover, the cohesiveness decreased when increasing the proportion of camel milk in the formulation. Also, the study of Gemechu et al. [33] recommended mixing cow milk with camel milk to obtain a good coagulum. Shahein et al. [12] recorded that the 10% addition of buffalo milk to camel milk improved curd hardness.

On the other hand, according to Arain et al. [44] and Jrad et al. [45], the use of mixed starter culture with camel chymosin improves the firmness and texture of camel milk coagulum.

From our results, it is important to point out that the textural properties of the fresh cheeses depend on the type of milk mixture, the type of coagulant, and the starter culture used.

### 3.6. Fourier Transform Infrared Spectroscopy Measurements

FTIR spectra recorded for CC cheese made with camel–goat milk mixture and coagulated with FDCR and MC cheese coagulated with FDMR were illustrated in Figure 2. The FTIR spectra of fresh cheeses (4000–400 cm^−1^) present a series of bands of different intensities.

The 3100–600 cm^−1^ spectral region was characterized by the presence of bands attributed to the O-H stretching vibrations of hydroxyl groups. Moreover, CC and MC fresh cheeses contain high percentages of water (65.01 and 64.23%, respectively), appearing as strong bands around 3344 and 3275 cm^−1^. This region refers to O-H and N-H stretching vibrations of hydroxyl groups and amide and amino acids and water bands that can cause masking of the band of N-H in CC and MC spectra; which should be presented in the 3330–3060 cm^−1^ region [46].

In addition, the peaks recorded in the 3100–2800 cm^−1^ spectral region are attributed to the C-H symmetric and asymmetric stretching vibrations of fatty acids [47]. Figure 2 showed two peaks in weak bands of CC (2850 and 2918 cm^−1^) and MC (2850 and 2919 cm^−1^) cheeses with different intensities. This could be attributed to the fat content of the cheeses studied [48].

The spectral region extending from 1800 cm^−1^ to 1600 cm^−1^ represents the C=O of acids and esters and shows the variations in the wavenumber range of 1741–1628 cm^−1^ for CC cheese and of 1741–1637 cm^−1^ for MC cheese. These results showed that both cheeses are fresh and that the level of lipolysis is low compared to the presence of ester bands in the fat.

The 1390–600 cm^−1^ spectral region is characterized by the presence of the Amide I and Amide II bands related to proteins. The Amide I is characterized by C=O and C-N stretching vibrations and the N-H bending vibration detected at 1542 and 1637 cm^−1^ for CC and MC cheeses, respectively. The Amide II is characterized by the stretching vibrations of C = N, N-H, and C-N functional groups and bending vibration of N-H group that is observed at ~1394 cm^−1^ for the CC cheese. For the MC cheese, no peak was recorded for Amide I. In addition, the 1390–1200 cm^−1^ region was characterized by peaks related to esters and aliphatic chains of fatty acids. In this region, a single peak was detected at 1238 cm^−1^ for the CC cheese.

The 1800–1200 cm^−1^ spectral region was characterized by the presence of bands assigned to C-C and C=O vibrations. Foda et al. [46] reported that these bands contain peaks related to polypeptides, amino acid, carbonyl groups of fatty acids, hydroxyl groups, carboxylic acid groups, and fatty acid esters (typically short chain) and represent alcohols, organic acids, and small water-soluble peptides. The results of the present study showed two peaks detected at 1174 and 1103 cm^−1^ for the CC cheese and two more peaks at 1160 and 1096 cm^−1^ for the MC cheese. This band is important in the analysis of the flavor of cheeses. Indeed, CC and MC cheeses do not contain several components responsible for flavor. This could be explained by the fact that both cheeses are fresh and contain the same type of starter culture.

### 3.7. X-Ray Diffraction Measurements

X-ray diffraction was carried out to evaluate the amorphous structure of cheese coagulated with FDCR (Figure 3). The X-ray diffraction pattern of CC and MC cheeses was amorphous. Indeed, it consisted of a very broad and diffuse line, indicating the absence of a structural unit that would be repeated identically at periodic intervals in three dimensions. In this context, it can be noted that the use of FDCR as a natural extract in the coagulation of camel–goat milk mixture did not lead to the appearance of a crystalline structure in the cheese obtained. These results assisted in understanding that the method used for the extraction of camel rennet did not lead to the appearance of nanoparticles retained in the CC cheese. These observations also recorded in the case of the MC cheese.

### 3.8. Microstructure of Fresh Cheeses

Figure 4 presents the microstructure of CC and MC fresh cheeses observed under an environmental scanning electron microscope (ESEM). From the micrographs, it appears that the protein network of caseins in CC cheese was more cohesive than that of MC cheese. In addition, the casein network is continuous for both types of cheese. The microstructure of CC cheese appears to have more and larger pores than those of MC cheese. Following the processing carried out on the two micrographs by the image J software, the number of holes was 195 for CC cheese and 163 for MC cheese over the entire micrographic area.

On the other hand, the casein network strands of the MC cheese are wider and thicker than those of the CC cheese. It also appears that the micrographic surface of CC cheese is smooth compared to that of MC cheese which is rough. These observations showed that the nature of the coagulating agent affects the texture of cheeses made with camel and goat-milk mixture. In addition, the use of FDCR extract in the coagulation of this mixture seems appealing to obtain a good coagulum.

Soltani et al. [49] reported that the type and concentration of coagulant had an important effect on the porous structure of the casein networks in the cheeses. In addition, Elnemr et al. [41] studied cheeses made from camel milk and coagulated with recombinant camel chymosin. These authors reported that the cheese microstructure was characterized by strands of thin aggregates, homogeneous structures, and continuous networks. Otherwise, Bouras et al. [16] reported that cheese made with a camel–goat milk mixture appears to have more numerous and smaller pores with a continuous and compact protein network. Compared with the results of the present study, it can be deduced that the use of FDCR had no significant effect on microstructural characteristics and more particularly on the number and size of pores and the appearance of the protein network. Abdalla et al. [13] recorded the absence of pores in the microstructure of cheeses made from milk mixed with just 10% and 30% of camel milk. Compared with the observed findings, the addition of 50% (*v*/*v*) goat milk to camel milk seems to have a relationship with the appearance of pores at the microstructural scale.

Regarding the texture profile, and according to our results, it is important to highlight that the number and diameter of the pores observed with the width of the strands of the casein network contributed to the difference in the texture parameters, mainly cohesiveness, adhesiveness, and chewiness. In this context, Bouazizi et al. [50] reported that protein aggregation is associated with springiness. Lower values of cohesiveness and adhesiveness for dromedary skim milk cheeses were reported owing to the presence of large pores in cheese.

### 3.9. Sensory Evaluation of Fresh Cheeses

The sensory profiles of CC and MC fresh cheeses are shown in Figure 4. Sensory attributes (smoothness, spreadability, creaminess, and texture) were significantly (*p* < 0.05) distinguished by the panel members and differed between CC and MC cheeses. The smell and taste attributes showed no significant differences between the two cheeses. CC and MC cheeses were, respectively, characterized by a moderately lactic odor (8 ± 0.3 and 9.23 ± 0.60), acid taste (7.12 ± 1.3 and 8.36 ± 0.9) and sweet taste (9 ± 0.4 and 10 ± 1.1).

In comparison with Bouras et al. [16], a less smooth (4.25 ± 1.64), less spreadable (9.2 ± 2.66), and less creamy (6.3 ± 0.14) texture was observed. In addition, the CC cheese was more preferred than the MC cheese with a score of 9/15. From these results, it can be deduced that the use of FDCR and the mixture of mesophilic and thermophilic starter culture had an impact on the texture of CC cheese. In addition, Bekele et al. [28] reported that camel milk cheese is characterized by a highly intense taste and aroma with a score 4.91/7 and 5.08/7, respectively. This sensory variation is justified by the difference between the lactic strains used in the manufacture of camel milk cheese. Furthermore, Sulieman et al. [51] noted that the use of a starter culture for the manufacture of white cheese with a camel–cow milk mixture had a clear influence on the sensory profile.

Additionally, microstructural differences could affect textural characteristics including the spreadability, smoothness, and creaminess of CC and MC cheeses.

## 4. Conclusions

This study evaluated the possibility of the use of freeze-dried camel rennet extract in the cheese-making process of camel–goat milk mixtures. Indeed, the amount of FDCR used for the coagulation process has been studied. The fresh cheese produced presented an acceptable cheese yield with good protein retention. Furthermore, the incorporation of the mixture of mesophilic and thermophilic starter cultures with FDCR resulted in a cheese with specific textural and microstructural properties; compared to the cheese made with FDMC. The obtained results highlighted the importance of producing cheeses from a mixture of camel and goat milk, which can help the industrialization of both milks to overcome particularly the problems related to the seasonality of camel milk production. Finally, further investigations should focus on the study of proteolysis during coagulation process to fully understand the mode of action of FDCR in comparison with FDMC.

## Figures and Tables

**Figure 1 foods-14-00334-f001:**
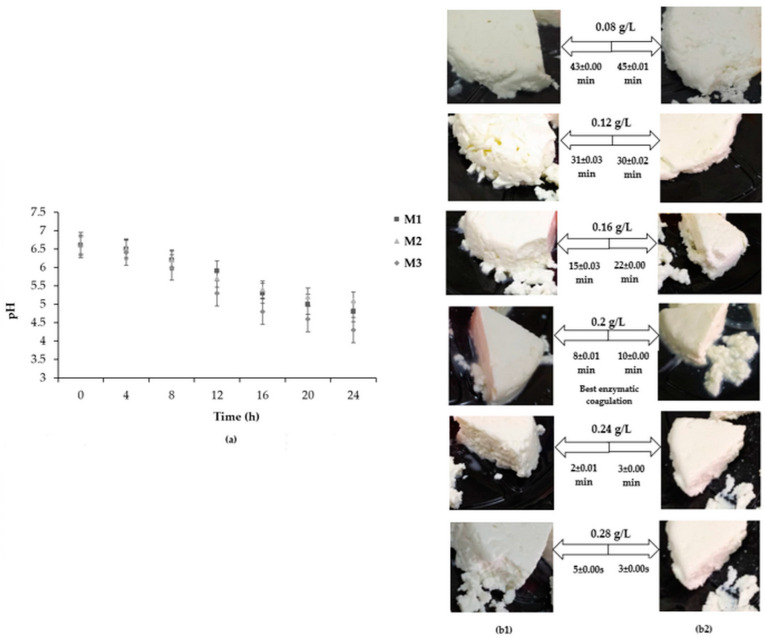
Acidification kinetic evolution of mesophilic starter culture (M_1_), thermophilic starter culture (M_2_), and a blend of mesophilic and thermophilic starter culture (M_3_) in a camel–goat milk mixture (**a**); Images of CC and MC fresh cheeses produced with different amounts of FDCR (**b1**) and FDMC (**b2**) during coagulation.

**Figure 2 foods-14-00334-f002:**
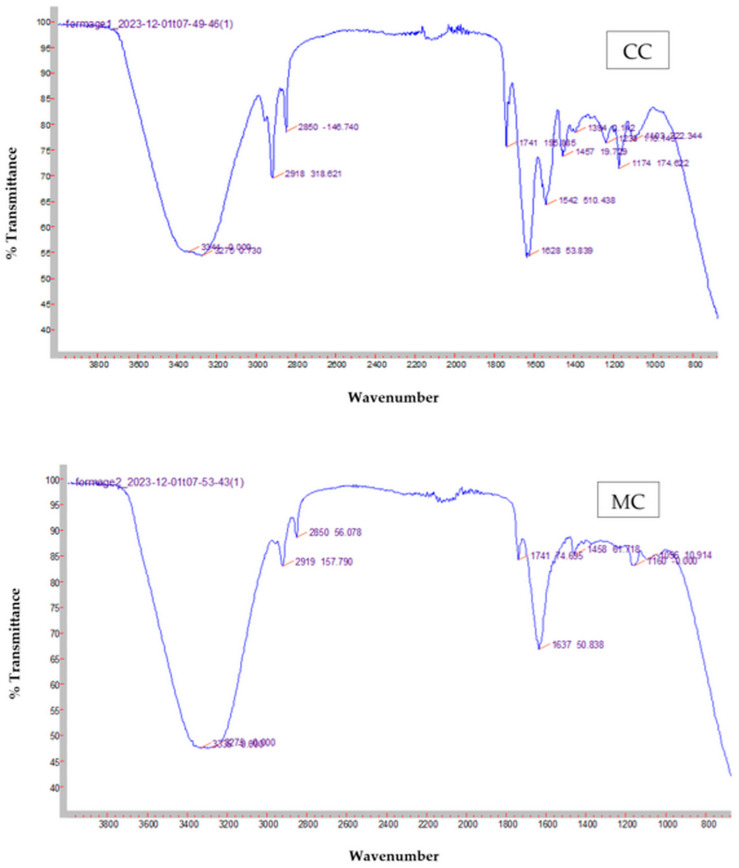
Average absorbance of the FTIR spectrum recorded for CC and MC fresh cheeses.

**Figure 3 foods-14-00334-f003:**
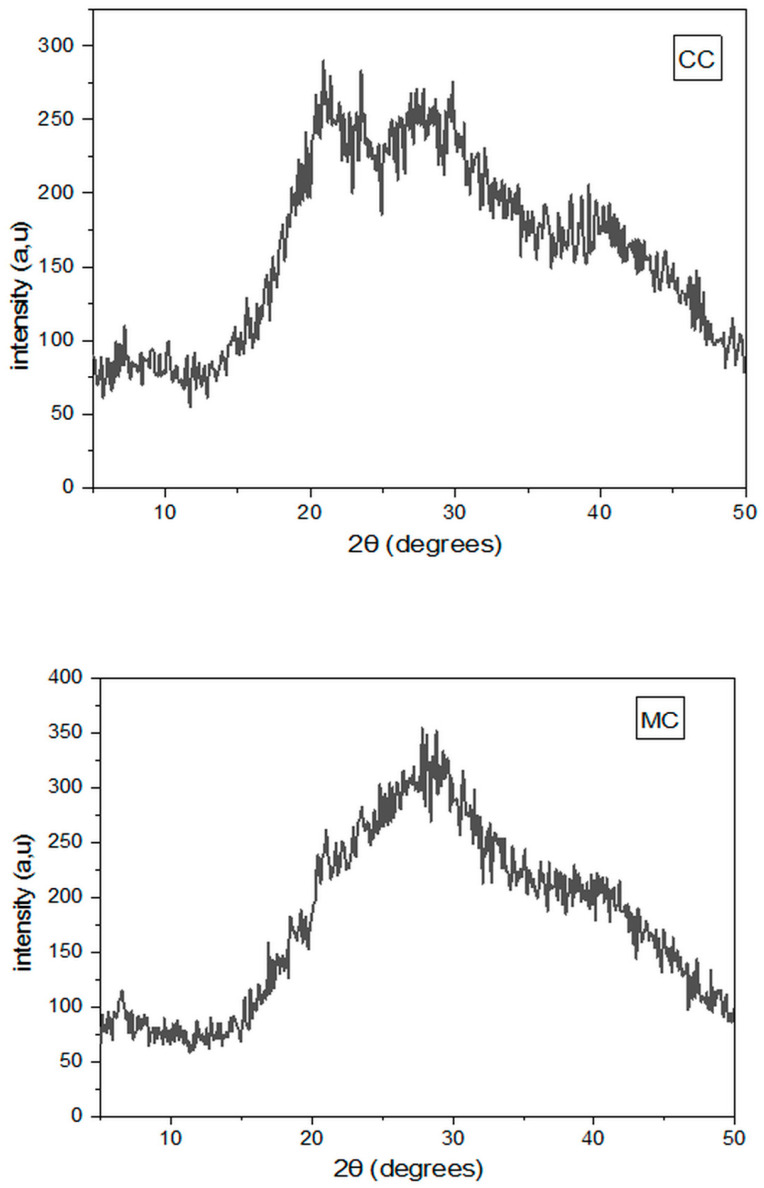
X-ray Diffraction spectrum recorded for CC and MC fresh cheeses.

**Figure 4 foods-14-00334-f004:**
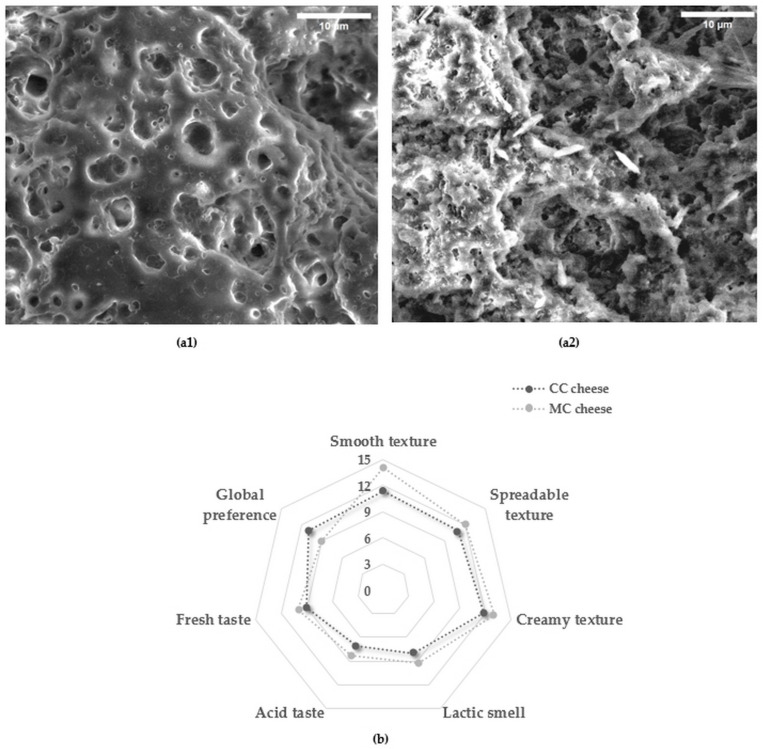
Scanning electron micrographs of CC (**a1**) and MC (**a2**) fresh cheeses at a scale of 10 µm and sensory properties (**b**).

**Table 1 foods-14-00334-t001:** Cheese yield values according to the added amount of FDCR and FDMC in the camel–goat milk mixture.

FDCR/FDMC (g/L)	Cheese Yield (%)		
	CY_1_	CY_2_	*p-*Value
0.08	16.58 ± 0.20	9.79 ± 0.14	*
0.12	15.88 ± 0.12	14.04 ± 0.12	*
0.16	19.28 ± 0.15	14.44 ± 0.12	*
0.20	16.50 ± 0.09	14.92 ± 0.07	*
0.24	19.01 ± 0.13	12.95 ± 0.20	*
0.28	16.87 ± 0.11	10.00 ± 0.15	*

FDCR: freeze-dried camel rennet, FDMC: freeze-dried microbial coagulant, CY_1_: cheese yield of fresh cheese made with FDCR, CY_2_: cheese yield of fresh cheese made with FDMC, * significant difference at *p* < 0.05.

**Table 2 foods-14-00334-t002:** Crude and mineral composition, color parameters and texture profile analysis of CC and MC finished fresh cheeses.

Parameters	CC Cheese	MC Cheese	*p-*Value
Physicochemical properties			
pH	4.50 ± 0.00	4.66 ± 0.00	*
a_w_	0.96 ± 0.00	0.97 ± 0.00	*
Dry matter (DM) (%)	34.99 ± 0.57	35.77 ± 0.18	ns
Protein (%/DM)	36.26 ± 1.75	27.20 ± 2.05	*
Fat (%/DM)	57.15 ± 1.05	67.09 ± 0.94	*
MFFB (%)	81.94 ± 1.31	84.51 ± 0.80	ns
Yield (%)	15.96 ± 0.10	13.55 ± 0.01	*
Ashes (g/100 g)	1.41 ± 0.02	3.55 ± 0.00	*
Ca^2+^ (mg/100 g)	4.33 ± 0.02	1.33 ± 0.01	*
P (mg/100 g)	2.86 ± 0.00	2.90 ± 0.02	ns
Na^+^ (mg/100 g)	123.66 ± 0.01	128.66 ± 0.00	*
K^+^ (mg/100 g)	73 ± 0.02	98.33 ± 0.00	*
Mg^2+^ (mg/100 g)	3 ± 0.00	1.33 ± 0.01	*
Fe (mg/100 g)	0.13 ± 0.00	0.13 ± 0.00	ns
Mn (mg/100 g)	<0.06	<0.06	ns
Cu (mg/100 g)	<0.03	<0.03	ns
Color parameters			
*L**	92.86 ± 0.23	93.97 ± 0.03	*
*a**	−1.03 ± 0.02	−1.75 ± 0.01	*
*b**	11.75 ± 0.00	10.26 ± 0.02	*
*C**	11.78 ± 0.08	10.33 ± 0.11	*
*h°*	95.04 ± 0.11	99.70 ± 0.04	*
Textural properties			
Hardness (N)	1.52 ± 0.32	1.20 ± 0.03	ns
Cohesiveness (−)	0.32 ± 0.01	0.67 ± 0.00	*
Springiness (mm)	14.25 ± 0.63	18.16 ± 0.63	*
Adhesiveness (N)	0.49 ± 0.07	0.81 ± 0.02	*
Chewiness (Nmm)	7.12 ± 1.44	14.80 ± 1.04	*

CC: fresh cheese coagulated with FDCR, MC: fresh cheese coagulated with FDMR; MFFB: moisture content on fat free basis. * significant difference at *p* < 0.05; ns: non-significant.

## Data Availability

The original contributions presented in this study are included in the article. Further inquiries can be directed to the corresponding authors.

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
