# Peer review of "Effect of Freeze-Dried Camel Rennet Extract on Coagulation of Camel–Goat Milk Mixture and Characterization of the Cheese Obtained"

_foods, 2025, doi:10.3390/foods14030334_

Round 1
Reviewer 1 Report
Comments and Suggestions for Authors
The study investigates the use of freeze-dried camel rennet extract (FDCR) as a coagulating agent for a mixture of camel and goat milk, comparing its performance with that of a microbial coagulant (FDMC). The authors evaluated the physicochemical properties, texture, microstructure, and sensory characteristics of the cheese produced. It is a relevant topic in dairy research, with a unique focus on camel-goat milk mixtures. However, more emphasis on novelty, detailed discussions, and methodological clarity will be needed.
1. Line 91 - 93: Ethical clearance for slaughterhouse practices should be mentioned.
2. Line 139 - 140: While coagulation times are optimized, the visual criteria for "tight coagulum without breakage" could benefit from more quantitative description.
3. Line 347 - 361: The results on cohesiveness and springiness are good, but it would be better to have more comparisons with other cheese types and literature benchmarks.
4. Line 430 - 434: Please revise the citation format, it should be XXX et. al [number] reported ... instead of just listed the citation number.
5. Line 421 - 443: While the use of SEM is good, there’s limited discussion on how the observed differences relate to sensory or textural properties.
6. Line 233 - 235: Please mention if post hoc tests were applied for pairwise comparisons and justify the use of Student's t-test.
Comments on the Quality of English LanguageThis manuscript is generally understandable. Some minor revisions can be made to improve clarity and improve grammatical precision.
1. Line 32: "The panelists were more preferred the cheese coagulated with FDCR." Correct to: "The panelists preferred the cheese coagulated with FDCR."
2. Line 70: "Freeze-dried rennet extract were used..." should be corrected to "Freeze-dried rennet extracts were used..."
3. Line 140: Consider rephrasing "tight coagulum without breakage" to "a firm and unbroken coagulum" for better readability.
4. Line 199: Clarify technical terms such as "FTIR spectrum" for general readers.
5. Line 155 - 156: Correct the formatting error in “FDCR/FDMC.After coagulation.”
6. Line 233 - 235: Mention if post hoc tests were applied for pairwise comparisons and justify the use of Student's t-test.
Reviewer 2 Report
Comments and Suggestions for Authors
Comments to the authors.
The aim of this study comparing two coagulation methods in the process of making fresh cheeses from a mixture of camel milk and goat milk.
The methodologies for evaluating the variables correspond to the type of study that was determined to be evaluated. They are well described and referenced.
There is a bit of confusion in the nomenclature of the treatments, I suggest that it be specified more clearly in the description of the methodology. Perhaps if it is included in the graphic summary it could be made clear or include a diagram of the cheese elaborated process.
In the description of the sensory methodology, indicate whether a trained or semi-trained panel was used and the duration of the training process.
Reviewer 3 Report
Comments and Suggestions for Authors
The manuscript contains several low-level errors of a non-academic nature, which need to be carefully corrected by the author. In particular, the quality of the figures (including their resolution and aesthetics) needs to be greatly improved. The author should fully recognize the key role that high-quality figures play in the publication of scientific papers.The following are specific comments for this article:
1. Line27. The author mentioned that the obtained cheese had fewer and less wide pores. Did the authors conduct a statistical analysis of the holes in the cheese and were the results significant?
2. The author uses a reference instead of the subject of the sentence. Please check whether this expression complies with the journal's requirements.
3. The author uses references instead of the subject of the sentence. Please check whether this expression meets the requirements of the journal. For example: Line 57, line 59, line 63
4. Line147. “were sieved at 50-µm”.This sentence lacks a subject.
5. Line154. There are two periods in one sentence.
6. Figure 1a is not standard and aesthetically pleasing and needs to be redrawn. In addition, Figure 1b is not clear enough, the text on the red and yellow labels in the figure is not clear, and the author did not explain the meaning of these two color labels.
7. Some of the text in the article is in small font size. Such as the ine255-256,line342-345 and line431-433.
8. The authors mentioned that adding goat milk to camel milk would change the acidification rate of the mixed milk, but did not analyze the possible reasons for this result.
9. In some of the results and analysis section, for example in the section of 3.1, 3.6 and 3.7.the authors usually just state the results without conducting an in-depth analysis of the results.
10. The resolution of Figure 2 is too low, and the numbers on the curve overlap a lot, making it difficult to obtain effective information.
11. The resolution of Figure 3 is too low, and even the numbers on the coordinate axes cannot be readable clearly.
12. What is the essential difference between FDCR and FDMC? In other words, what is the difference in the ingredients and composition of these two coagulating agent?
13. Why did the author perform X-ray analysis on cheese? What practical significance did the X-ray test results revealed?
14. The SEM image is not clear. It is recommended to conduct quantitative analysis of the SEM image, such as a statistical analysis of the holes' number per unit area of cheese obtained under different conditions.
Round 2
Reviewer 3 Report
Comments and Suggestions for Authors
The comments and questions raised by the reviewer have been addressed in detail by the authors. After the authors' careful revisions, this version of the manuscript has improved significantly compared to the previous one.
The authors conducted a comparative study on the effects of two different coagulating agents on cheese made from camel-goat milk 23 mixture. The experimental design is rational, and the results reveal some practical significance. However, there is still much room for improvement in the quality of the figures and tables, and the reviewer hopes that the authors will do their best to refine them.
